# The Protective Effect of Xanthohumol on the Content of Selected Elements in the Bone Tissue for Exposed Japanese Quails to TCDD

**DOI:** 10.3390/ijerph17165883

**Published:** 2020-08-13

**Authors:** Aleksandra Całkosińska, Marzena Dominiak, Sylwia Sobolewska, Anna Leśków, Małgorzata Tarnowska, Aleksander Całkosiński, Maciej Dobrzyński

**Affiliations:** 1Department of Oral Surgery, Wroclaw Medical University, Krakowska 26 Street, 50-425 Wroclaw, Poland; marzena.dominiak@umed.wroc.pl; 2Department of Animal Nutrition and Feed Management, Wroclaw University of Environmental and Life Sciences, Chelmonskiego 38c Street, 51-630 Wroclaw, Poland; sylwia.maria.sobolewska@gmail.com; 3Department of Nervous System Diseases, Faculty of Health Science, Wroclaw Medical University, Bartla Street 5, 51-618 Wroclaw, Poland; anna.leskow@umed.wroc.pl (A.L.); malgorzata.tarnowska@umed.wroc.pl (M.T.); 4Students’ Scientific Association of Biomaterials and Experimental Dentistry, Wroclaw Medical University, Bujwida 44 Street, 50-368 Wroclaw, Poland; olekcal4@gmail.com; 5Department of Conservative Dentistry and Pedodontics, Wroclaw Medical University, Krakowska 26 Street, 50-425 Wroclaw, Poland; maciej.dobrzynski@umed.wroc.pl

**Keywords:** TCDD, dioxins, antioxidants, xanthohumol, bone mineral composition, bone mineralization

## Abstract

Dioxins (including 2,3,7,8-tetrachlorodibenzo-*p*-dioxin (TCDD) are highly toxic and persistent chemicals widely distributed in the environment in trace amounts, and are side products of industrial and chemical processes. Exposure to dioxins leads to multiorgan morphological and functional abnormalities, including within the bone tissue, disrupting its microarchitecture and mechanical properties. Xanthohumol (XN) is a chemical compound classified as a prenylated flavonoid, distinguished by multidirectional biological action. The aim of the study is to assess whether xanthohumol, as a substance with strong antioxidant and anti-inflammatory properties, has the ability to eliminate the negative effects of TCDD on bone tissue. The experiment was conducted on adult Japanese quails. Two different doses of TCDD and xanthohumol were administered to birds. After euthanasia of animals, the research material in the form of cranial vault and hind limb bone was collected, and their mineral compositions of calcium, phosphorus, magnesium, zinc, and iron concentrations were determined using atomic emission spectrometry in an acetylene-air flame method. Our results indicate that the administration of TCDD at a low dose causes more dynamic changes in the concentration of elements in bone, in comparison to a higher dose of dioxin. Results show also that higher doses of the XN cause the linear increase in the concentration of phosphorus and iron in the bone of the hind limb, and calcium in the bones of the cranial vault. In conclusion, our experiment shows that the use of TCDD and XN in Japanese quails together in various doses influences the content of phosphorus, magnesium, zinc, and iron in the research material.

## 1. Introduction

This research was undertaken to determine the effect of xanthohumol (XN) on biological effects caused by 2,3,7,8-tetrachlorodibenzo*-p*-dioxin (TCDD) in bone tissue. Dioxins are substances found in the environment in negligible amounts, formed as byproducts during industrial processes, such as production of herbicides, pesticides, and waste incineration [1,2,3]. Dioxins belong to the group of persistent organic compounds, and accumulate in the environment at every stage of the food chain. It leads to their accumulation in the adipose tissue of organisms, and permanent exposure to low doses of dioxins, which may result in the occurrence of various types of health disorders. In places where ecological disasters have occurred (e.g., Seveso (Italy)), the population is particularly exposed to the effects of dioxins. This justifies the search for pharmacological agents that have protective effects against dioxins. The best-known substance that represents this group is TCDD, which is classified as one of the most potent toxins. The molecular activity of dioxins is based on constant activation of the aryl hydrocarbon receptor (AhR), which, for example, induces the transcription of genes of different molecular forms of cytochrome P-450 (CYP), an enzyme responsible for metabolism of xenobiotics [4,5,6]. The exposure of organisms to dioxins causes numerous multi-organ disorders, for example, cancer progression, chloracne, liver enzymes overproduction, metabolism disruption, diabetes, reproductive disorders, thyroid function disorders, cardiovascular diseases, and also abnormalities of bone tissue, disrupting its microarchitecture and mechanical properties [7,8,9,10,11,12,13,14]. Based on the outcome of the experimental research, the TCDD has been shown to induce the oxidation stress, which causes the increase in the concentration of proinflammatory interleukins and activation of the osteoclastogenesis [15,16,17]. Moreover, the TCDD inhibits activity of alkaline phosphatase and the other initiators of mineralization. That can result in formation of hard tissues with low mineral levels [17,18]. The bones of the cranial vault and long bones differ in their histological structure, which is associated with their slightly various functions [4,19,20,21,22]. The intensity of bone metabolic processes is associated with the availability of transmission of signals—and therefore, with blood flow [23]. The bones are subjected to constant mechanical loads, and due to that, changes in the blood pressure and flow [24].

Bone remodeling is the key process for maintaining the required bone histophysiology, which depends on the activity of bone cells [25]. The disruption of this process may be associated with bone resorption abnormality associated with osteoblast activation, or may interfere with the activity of the osteoblasts responsible for the synthesis and mineralization of the bone and the osteoclast activation.

Xanthohumol (XN) is a bioactive substance obtained from female hop inflorescences (*Humulus lupulus* L.) with a range of biological properties [26]. The action of the XN on bone tissue is based mainly on the strong inhibition of bone resorption and the induction of osteogenesis. Depending on the dose, the XN stimulates the expression of osteogenic marker genes (Runx2, ALPL, and BGLAP), as well as alkaline phosphatase (ALP) activity in mesenchymal and preosteoblastic mouse cell lines [27]. The recent studies have shown that the XN significantly inhibits the bone resorption induced by NF-κB receptor activator ligand (RANKL) [28]. Xanthohumol can neutralize the effect of dioxins potentiating irregularities in the bone tissue related with osteoporosis. In the study of Gao et al. [29], the XN was shown to suppress the activity of T lymphocyte proliferation and cytokine production by Th1 lymphocytes (IL-2, IFN-γ and TNF-α). Activated T cells and the cytokines activated by them lead to RANKL expression on osteoblasts. In addition, stimulated T lymphocytes directly produce RANKL, which, by activating a specific RANK receptor, induces the formation and activation of osteoclasts [30]. In addition, it has been observed that xanthohumol can stimulate osteoblast differentiation, induce alkaline phosphatase (ALP) activity, and increase marker gene expression of RUNX2 osteoblasts in mouse MC3T3-E116 cells [31].

The XN as a phytoestrogen has the effect of inhibition of the differentiation and recruitment of osteoclasts; in this way it can inhibit the resorption process, causing the increase of the calcium content in bone [32]. On the other hand, it enhances the proliferation of various body cells, including osteoblasts [33].

Differently, in natural conditions, osteoblasts synthesize the bone matrix and then induce its mineralization. Presumably, a double dose of XN increased the proliferation of osteoblasts, which expanded the bone’s matrix volume without leading to its mineralization—this leads to decreased content of the calcium.

The described action of the XN was observed by Suh K.S. et al. in the paper about the impact of the XN on osteoblast differentiation [34]. This may mean that XN may have a multidirectional effect on the calcium content in the bone tissue by reducing its resorption or increasing osteoblast differentiation, depending on the adjustment. Available in vitro studies explain this mechanism only partially [33,34].

It was proven that administration of the substances, such as α-tocopherol, a powerful antioxidant antagonist of AhR, and acetylsalicylic acid, prevent the destructive influence of the TCDD on the bone tissue [35,36,37,38,39,40].

Based on evidence confirming the anti-inflammatory and antioxidant activity of xanthohumol, it can be assumed that this substance stimulates bone healing and has a protective effect against chemical damage associated with dioxin exposure [27,31,41].

The aim of this experiment was to determine the effect of xanthohumol on the bone tissue of organism exposed to dioxins.

## 2. Materials and Methods

### 2.1. Animals and Experimental Groups

The experimentation, transportation, and care of the animals were performed in compliance with the relevant laws and the institutional guidelines. The consent of the 1st Local Ethical Commission for Animal Experiments was obtained at the Institute of Immunology and Experimental Therapy of the Polish Academy of Sciences in Wroclaw, Weigla Street 12, 53-114 Wroclaw (consent number: 52/2015), before conduction of the experiment. 

The experiment was performed on adult, female Japanese quails (*Coturnix japonica*) with a body weight of ca. 160 g. Birds were assigned to experimental groups by randomization. Japanese quails were chosen as an experimental model because of their significant sensitivity to dioxins [42].

Animals were kept in four-story metal cages, where the following conditions were provided: 15 air changes per hour, average temperature 25 °C, average humidity around 47%, light cycle 12/12 h. The temperature and humidity were measured each day of the experiment. Three or four birds were kept in each cage, due to the herd character of this species. Japanese quails were provided with constant access to water and fed ad libitum with complete mixtures. With constant access to feed, the herd hierarchy did not affect the animals’ ability to feed.

Birds were randomly assigned to nine groups, each with seven individuals (Table 1).

The experiment lasted 21 days. Feed intake was recorded daily. All quails were weighed on the first and the last day of the experiment. On the first day of the experiment, TCDD was administered to birds intramuscularly in two different doses (0.5 μg/kg body weight (b.w.) and 2 μg/kg b.w.) in groups IV, V, VI, VII, VIII, and IX. Xanthohumol was administered to animals by intramuscular injection the first, seventh, and fourteenth days of experiment at a dose of 10 mg/kg b.w. and 20 mg/kg b.w. in groups II, III, V, VI, VIII, and IX.

Quails were euthanized after 21 days from the beginning of experiment. Tested tissue in the form of bone of cranial calvaria and hind limb was collected from animals. Obtained samples were placed in sterile containers and frozen at the temperature of −73 °C.

### 2.2. Chemical Substances Used in the Experiment

The following substances were used in the study: xanthohumol (Department of Chemistry, Faculty of Biotechnology and Food Sciences, Wroclaw University of Environmental and Life Sciences); 2,3,7,8-tetrachlorodibenzo-*p*-dioxin (TCDD) standard solution (Greyhound Chromatography and Allied Chemicals, No. DD-2378-S) dissolved in dimethyl sulfoxide (DMSO) at a concentration of 5 μg/mL was prepared in Department of Chemistry, Faculty of Biotechnology and Food Sciences, Wroclaw University of Environmental and Life Sciences; and thiopental (Biochemie GmbH, Vienna, Austria.

### 2.3. Determination of the Mineral Composition of the Cranial Vault and Hind Limb Bone of Examined Animals

The mineral composition of the cranial vault and hind limb bone was determined by assessing the concentration of calcium (Ca), phosphorus (P), magnesium (Mg), zinc (Zn), and iron (Fe).

#### 2.3.1. Mineralization of Bone Tissue Samples

The mineralization of samples was carried out in a closed, wet digestion microwave system. To the homogeneous sample of bone (from 0.1 g to 0.5 g), 5 cm^3^ of concentrated nitric acid (V) and 1 cm^3^ concentrated hydrogen peroxide were added. Then, the samples were mineralized in a MARS 5 microwave sample preparation system. Mineralizates were transferred to measuring vessels with the capacity 10 cm^3^, using redistilled water.

#### 2.3.2. Determination of the Concentration of Calcium, Phosphorus, Magnesium, Zinc, and Iron in the Skull Bone and Hind Limb of the Tested Birds

The calcium content was determined by the method of atomic emission spectrometry in an acetylene-air flame, using the SpectrAA atomic absorption spectrometer with a Varian AA240FS flame adapter. The phosphorus content was determined by spectrophotometry, using a Unicam UV300 spectrophotometer from ThermoSpectronic. The determination of zinc, iron, and magnesium content was carried out by the absorption spectrometry method in an acetylene-air flame, using the SpectrAA atomic absorption spectrometer with a Varian AA240FS flame adapter.

The concentrations of chemical elements in tested material were determined in accordance with the following standards: European Standard PN-EN 13805: 2014-11; calcium determination according to the Research Procedure PB-06/AAS; determination of phosphorus according to the Polish Standard PN-EN ISO 3946: 2000; magnesium determination according to the Polish Standard PN-EN 15505: 2009; determination of zinc and iron according to the Polish Standard PN-EN 14082: 2004.

#### 2.3.3. Statistical Analysis

The development of research results consisted of the calculation of basic descriptive statistics of elemental values (mean, standard deviation, median, quartiles, and extreme values). The Bartlett’s test was used to check the homogeneity of the variance of results. The significance of the effect of two controlled factors, TCDD and xanthohumol, on the content of the analyzed elements in the bone of cranial vault and hindlimb was verified on the basis of two-factor analysis of variance. Because the empirical distributions were significantly different from the normal distribution, the raw results were subjected to Box–Cox transformation. All hypotheses were verified at the significance level *p* ≤ 0.05. Statistical analysis was performed using the STATISTICS 13 PL.

## 3. Results

### 3.1. Calcium

The only factor that significantly affected the content of calcium in cranial vault bone was the dose of xanthohumol (*p* < 0.001). The dose of 10 mg/kg b.w. caused a significant increase in concentration of calcium. The increase of dose to 20 mg/kg b.w. reduced calcium levels (Figure 1a).

The only factor that significantly affected calcium content in the hind limb bone was the dose of TCDD (*p* < 0.05). The doses of 0.5 μg/kg b.w. and 2 μg/kg b.w. caused a significant increase in Ca concentration (Figure 1b).

### 3.2. Phosphorus

Both dioxin TCDD (*p* < 0.001) and xanthohumol (*p* < 0.001) proved to be factors that significantly reduced the phosphorus content in the bone of the cranial vault of the birds. No statistically significant correlation of these factors was observed (*p* > 0.05). Increasing the dose of xanthohumol caused a linear decrease in phosphorus content, while in the case of TCDD, even a dose of 0.5 μg per kilogram of body weight caused a sharp decrease in phosphorus concentration in the cranial vault bone (Figure 2a).

The change in phosphorus content in the bones of the hind limb was significantly influenced by the dose of xanthohumol (*p* < 0.01). A dose of xanthohumol 10 mg per kilogram of body weight caused a sharp increase in phosphorus content (*p* < 0.01), and its further increase had no significant effect. The effect of TCDD on the phosphorus content was small. Only a dose of 2 μg per kilogram of body weight caused a slight decrease in phosphorus concentration. A statistically significant correlation between the dose of xanthohumol and TCDD was also observed (*p* < 0.05) (Figure 2b).

### 3.3. Magnesium

The magnesium concentration in the bone of cranial vault depended significantly on both factors. There was also a statistically significant correlation between TCDD and xanthohumol. No statistically significant results were obtained for the magnesium content in the hind limb of the tested animals (Figure 3).

### 3.4. Zinc

The concentration of zinc in the bones of cranial vault depended significantly on both factors. There was also a statistically significant correlation between TCDD and xanthohumol. The content of this element in the bone of the hind limb significantly depended on the dose of dioxin. There was also a statistically significant correlation between TCDD and xanthohumol (Figure 4a,b).

### 3.5. Iron

The concentration of iron in the bones of cranial vault depended significantly on both factors. However, no statistically significant correlation was observed between the levels of TCDD and xanthohumol. The concentration of iron in the bones of the hind limb depended significantly on both factors. There was also a statistically significant correlation between TCDD and xanthohumol (Figure 5a,b).

## 4. Discussion

In previous studies, a negative effect of dioxins on the bone tissue, associated with their pro-inflammatory properties and ability to induce oxidative stress, has been demonstrated [9,15,43]. The purpose of the carried-out experiment was to determine the effect of XN on the bone tissue after administration of TCDD in tested animals. The indicator of this dependency is the mineral composition of cranial vault bone and hind limb of Japanese quail treated with TCDD and xanthohumol. The assessment of phosphorus, calcium, magnesium, zinc, and iron content makes it possible to determine the effectiveness of bone tissue mineralization processes in tested animals [44,45,46]. Calcium and phosphorus are the building materials of the calcium dihydroxyapatite crystal, which is the basic structural unit of bone tissue, forming an ordered crystal network undergoing constant physiological reconstruction. Magnesium, zinc, and iron are elements that play an important role in regulation of bone metabolism. Previous experimental studies have proven the adverse effects of dioxins as environmental factors on bone remodeling in the experimental animals, both in developmental and adult age [43]. It was found that exposure to TCDD led to decrease of bone mechanical strength and caused an increase in density of cortical bone in studied adult rats [4,19,47,48,49].

In the experiment conducted on animals during prenatal and lactation period, an adverse effect of dioxins on the size, strength, and mineralization of bones was observed [50]. Similar results were obtained, in their studies, by Miettinen et al. [11], exposing dioxin to female rats at various periods of pregnancy and lactation. In the offspring of groups exposed to TCDD, bone length, cortical cross-sectional area, and mineral density decreased. Mechanical tests showed significantly reduced tibial and femoral fracture strength, as well as their increased stiffness.

In Dobrzyński’s experiment [50], the reduction of calcium content in bone material, in the form of cranial vault and knee joint, was demonstrated in rat newborns whose mothers were given TCDD at a dose of 5 μg/kg b.w. Moreover, other studies found that the offspring of TCDD-treated mice have abnormalities in the size and shape of the jaw and cleft palate [51,52].

Our research carried out as part of this study confirms the effect of dioxins on the bone, resulting in disorders in its mineral composition. It was observed that the use of TCDD only at a dose of 0.5 μg/kg b.w. led to a sharp decrease in the concentrations of phosphorus, magnesium, zinc, and iron, while the dose of 2 μg/kg b.w. caused a significant reduction in the calcium content in the bone of the cranial vault of the tested Japanese quail. In the hind limb of animals exposed to a lower dose of dioxin, there was a decrease in zinc and iron, and—interestingly—a significant increase in calcium. According to Lind et al. [53], increased calcium content in the bone of rats exposed to TCDD can be associated with the disturbed mineralization of this tissue, and inhibition of osteocytic osteolysis, which is a characteristic feature for older animals and may negatively affect the bone’s mechanical properties.

Our research conducted as part of this experiment also shows that a higher dose of dioxin led to a significant decrease in the concentrations of phosphorus, calcium, and zinc. The bone’s mineral disorders may be associated with the induction of inflammation and oxidative stress by increasing the expression of proinflammatory cytokines and COX-2 [43,50], which may result in imbalance between bone mineralization and resorption in favor of catabolic processes.

The previous reports confirm the need to search for substances that eliminate or reduce the effects of dioxins on the bone tissue [4,19,48,50].

Dobrzyński et al. showed a beneficial effect of tocopherol and acetylsalicylic acid on the concentrations of calcium and magnesium in the bones of the cranial vault and knee joint, which was observed in the offspring of rats exposed to TCDD [54]. This confirms the important role of substances with anti-inflammatory and antioxidant properties in reducing the biological effects of dioxins.

The XN characterized by the above-described properties can neutralize the effects of dioxins potentiating irregularities in bone tissue. In studies performed on mice with ovariectomy-induced osteoporosis, the severity of bone resorption decreased. The XN administration to the animals was started four weeks after ovarian resection, and was treated for five weeks. The histomorphometric analysis of the lumbar vertebrae showed that trabecular bone volume was significantly higher in XN-treated mice. The results confirming the beneficial effects of prenylated flavonoids on bone resorption associated with osteoporosis were also obtained by Ban et al. [55]. The use of Lifenol^®^, which contains hops extracts, in ovariectomized rats significantly reduced the decrease in bone mineral density and mineral content. It has been observed that XN stimulates osteoblast differentiation by activating the RUNX2 transcription factor [56]. However, in the experiment performed on osteoblasts of the MC3T3-E1 line treated with TCDD, the level of protein kinase C and NF-κB increased. Nevertheless, the presence of XN alleviated the pathological effects of TCDD and led to an increase in gene expression associated with osteoblast differentiation, such as alkaline phosphatase, osteocalcin, and osteoprotegerin [34].

The eliminating effects of XN on the biological effects induced by TCDD administration may be related to the ability of prenylated flavonoids to block the activation reaction mediated by cytochrome P450, which was observed in studies carried out in various in vitro systems. In a study by Henderson et al. [57], the XN caused complete inhibition of resorufin O-deetylase (EROD) CYP1A1 and CYP1B1 at an appropriate concentration, while showing less effect on CYP2E1 and CYP3A4.

Our own research showed that the use of only XN at a dose of 10 mg/kg b.w. led to an increase in calcium content in the cranial vault bone.

The use of xanthohumol at a dose of 10 mg/kg b.w. and 20 mg/kg b.w. in animals subjected to dioxin exposure (0.5 μg/kg b.w.) led to a marked reduction in the dynamics of the decrease in magnesium and zinc content in the bone of the cranial vault, and an increase in phosphorus and zinc in the hind limb of tested animals.

When using dioxin at a dose of 2 μg/kg b.w. in combination with xanthohumol at a dose of 10 mg/kg or 20 mg/kg b.w., there was always substantial decrease in the content of individual mineral substances. This result may indicate that the used dose of xanthohumol was insufficient to eliminate the effect of dioxin on bone tissue at a dose of 2 μg/kg b.w. of TCDD in Japanese quails tested.

Nevertheless, it can be stated that xanthohumol limits the intensity of demineralization processes in the bone tissue induced by dioxin, and has osteoprotective effects.

## 5. Conclusions

The administration of 2,3,7,8-tetrachlorodibenzo-*p*-dioxin in Japanese quails at a low dose causes more dynamic changes in the concentration of elements in bone in comparison to a higher dose of dioxin. For most bioelements (P, Mg, Zn, Fe), there is a sharp decrease in their content noticed, except for calcium, whose concentration increased significantly.Higher doses of the XN cause the linear increase in the concentrations of phosphorus and iron in the bone of the hind limb, and calcium in the bones of cranial vault. Most often, increasing the dose of xanthohumol led to a linear decrease in the content of bioelements.The use of dioxin and xanthohumol in Japanese quails together in various doses influenced the content of phosphorus, magnesium, zinc, and iron in the research material.

## Figures and Tables

**Figure 1 ijerph-17-05883-f001:**
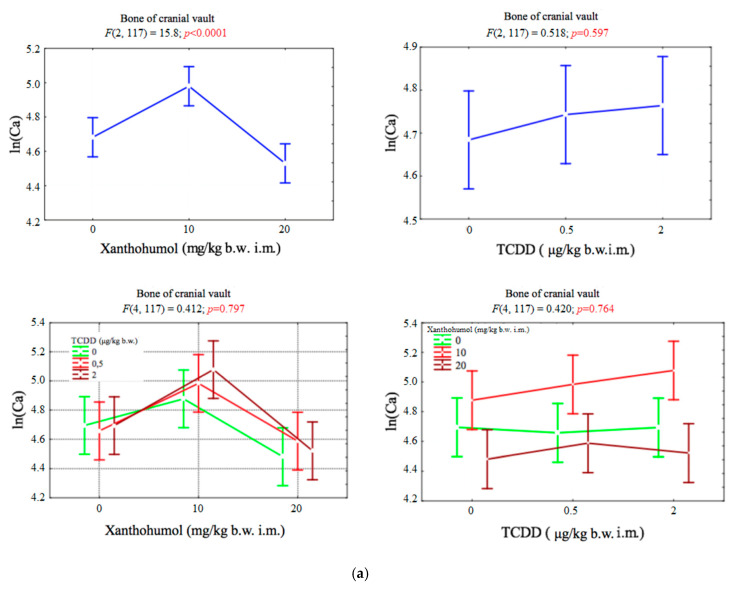
(**a**) Average calcium concentration (transformed variables) in the bone of the cranial vault, depending on the dose of TCDD and xanthohumol, and results of the significance test. Results are represented as mean ± SD. *p* ≤ 0.05 was considered statistically significant. Significant *p* value is highlighted in bold. (**b**) Average calcium concentration (transformed variables) in the bone of the hind limb, depending on the dose of TCDD and xanthohumol, and results of the significance test. Results are represented as mean ± SD. *p* ≤ 0.05 was considered statistically significant. Significant *p* value is highlighted in bold.

**Figure 2 ijerph-17-05883-f002:**
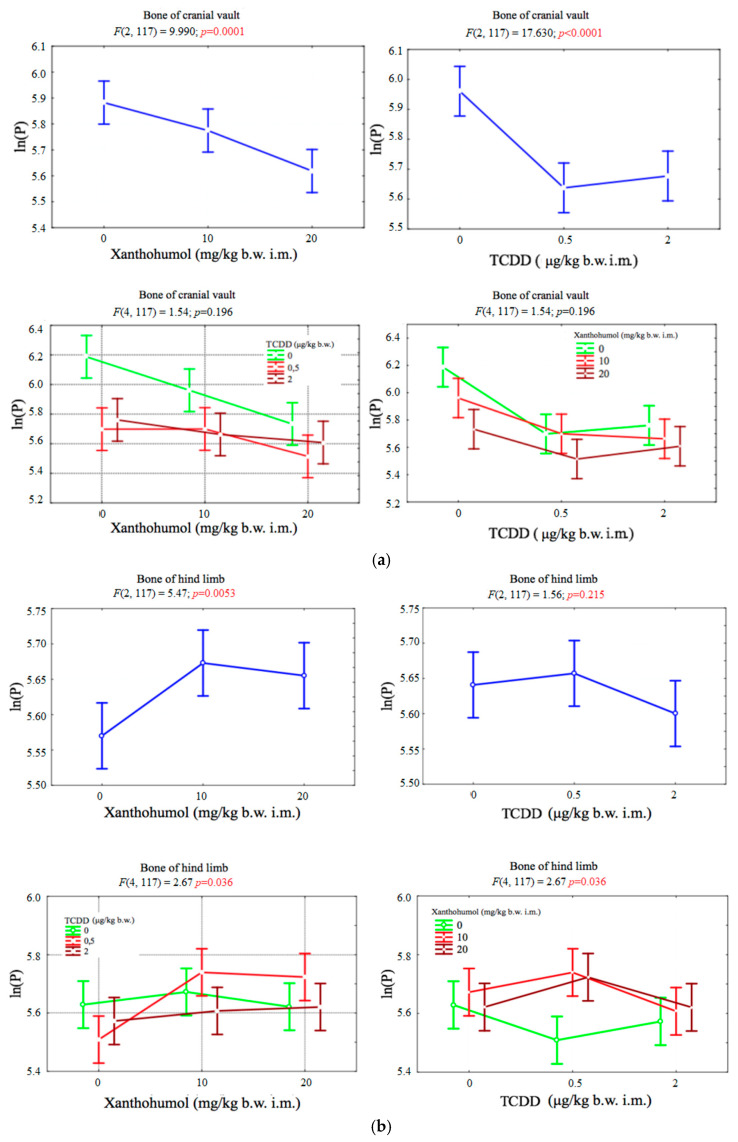
(**a**) Average phosphorus concentration (transformed variables) in the bone of the cranial vault, depending on the dose of TCDD and xanthohumol, and results of the significance test. Results are represented as mean ± SD. *p* ≤ 0.05 was considered statistically significant. Significant *p* value is highlighted in bold. (**b**) Average phosphorus concentration (transformed variables) in the bone of the hind limb, depending on the dose of TCDD and xanthohumol, and results of the significance test. Results are represented as mean ± SD. *p* ≤ 0.05 was considered statistically significant. Significant *p* value is highlighted in bold.

**Figure 3 ijerph-17-05883-f003:**
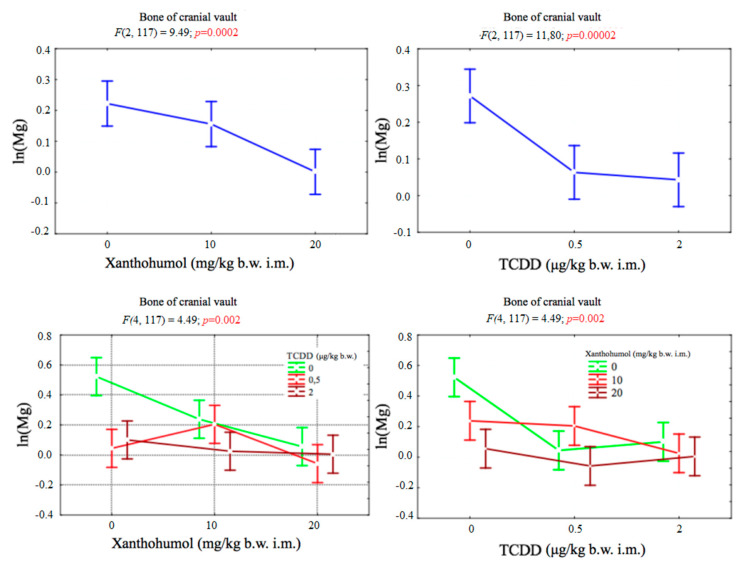
Average magnesium concentration (transformed variables) in the bone of the cranial vault, depending on the dose of TCDD and xanthohumol, and results of the significance test. Results are represented as mean ± SD. *p* ≤ 0.05 was considered statistically significant. Significant *p* value is highlighted in bold.

**Figure 4 ijerph-17-05883-f004:**
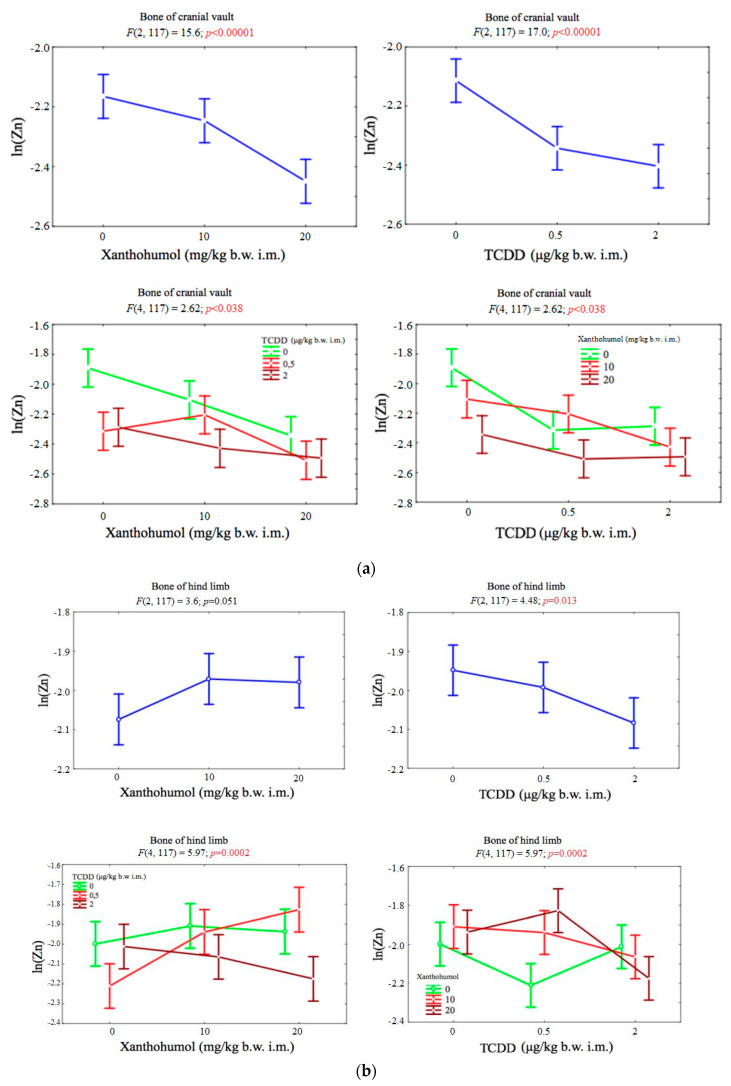
(**a**) Average zinc concentration (transformed variables) in the bone of the cranial vault, depending on the dose of TCDD and xanthohumol, and results of the significance test. Results are represented as mean ± SD. *p* ≤ 0.05 was considered statistically significant. Significant *p* value is highlighted in bold. (**b**) Average zinc concentration (transformed variables) in the bone of the hind limb, depending on the dose of TCDD and xanthohumol, and results of the significance test. Results are represented as mean ± SD. *p* ≤ 0.05 was considered statistically significant. Significant *p* value is highlighted in bold.

**Figure 5 ijerph-17-05883-f005:**
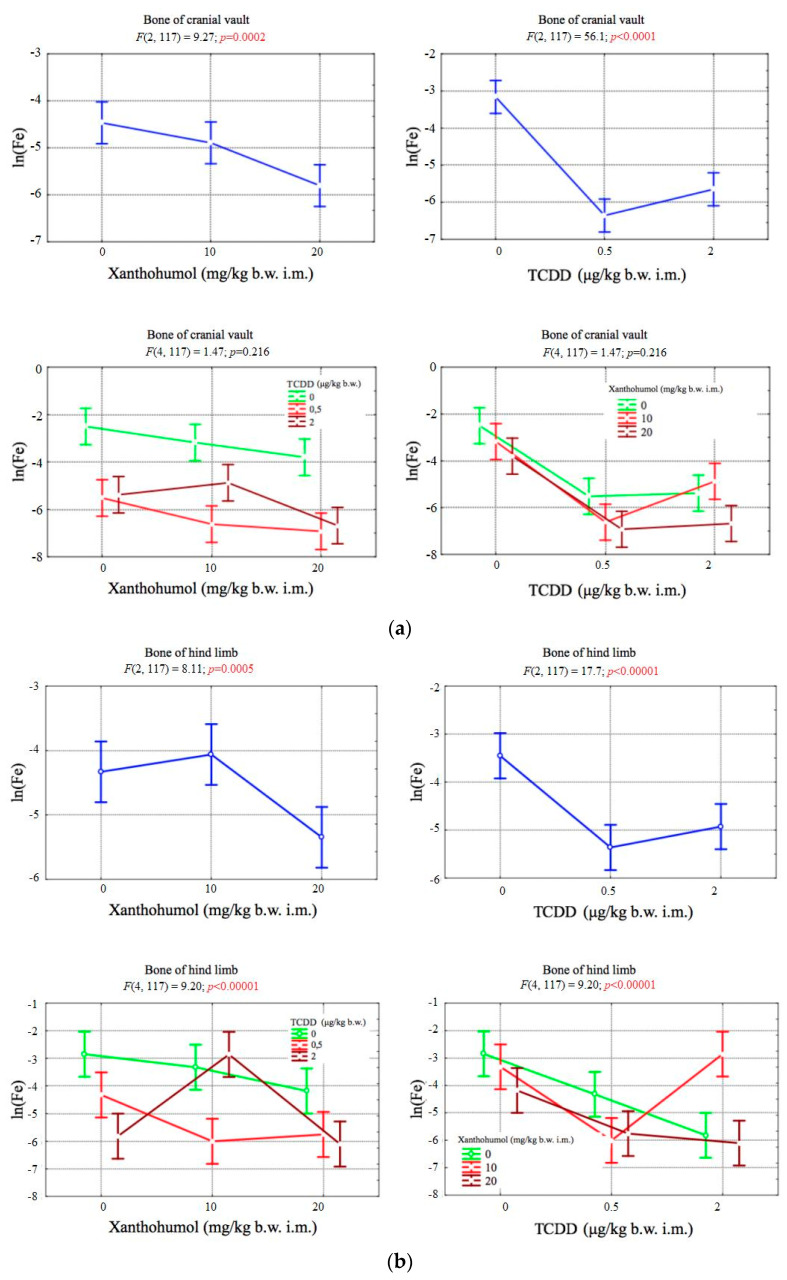
(**a**) Average iron concentration (transformed variables) in the bone of the cranial vault, depending on the dose of TCDD and xanthohumol, and results of the significance test. Results are represented as mean ± SD. *p* ≤ 0.05 was considered statistically significant. Significant *p* value is highlighted in bold. (**b**) Average iron concentration (transformed variables) in the bone of the hind limb, depending on the dose of TCDD and xanthohumol, and results of the significance test. Results are represented as mean ± SD. *p* ≤ 0.05 was considered statistically significant. Significant *p* value is highlighted in bold.

**Table 1 ijerph-17-05883-t001:** Scheme of the animal groups formed for the experiment. The animals were divided into nine groups, including a negative control group. 2,3,7,8-tetrachlorodibenzo-*p*-dioxin (TCDD) and Xanthohumol (XN) were administered intramuscularly to birds in different doses.

Group Number	TCDD (ug/kg b.w. i.m.)	Xanthohumol (mg/kg b.w. i.m.)
**I**	0	0
**II**	0	10
**III**	0	20
**IV**	0.5	0
**V**	0.5	10
**VI**	0.5	20
**VII**	2	0
**VIII**	2	10
**IX**	2	20

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
