# Peer review of "The Protective Effect of Xanthohumol on the Content of Selected Elements in the Bone Tissue for Exposed Japanese Quails to TCDD"

_ijerph, 2020, doi:10.3390/ijerph17165883_

Round 1
Reviewer 1 Report
The manuscript handles with the protective effect of xanthohumol on biological effects caused 2,3,7,8-tetrachlorodibenzo-p-dioxin (TCDD) in bone tissue, although contains some questions.
The introduction has to be concise and objective, but in this case, it starts with one objective, that is repeated 5 phrases after, which is a little strange for those who read. The introduction should give a framework of what is intended to be done and only at the end is the set the objective. In this work, the introduction section, seems to me, a set of random phrases with topics that will be covered and shorter than abstract. In my opinion, have to be reformulated and extended to cover all aspects that will be developed. For example, what kind of negative effects of TCDD can be promoted on bone tissue? What are the probable mechanism involved to decrease or eliminate these negative effects? Nobody else tried to try other compounds in order to get the same result and why xantohumol? Why were measured the concentrations “of calcium (Ca), phosphorus (P), magnesium (Mg), zinc (Zn) and iron (Fe)”?
Section 3.
In all figures change decimal divisor (,) by (.)
Section 4. Discussion
It needs references for the first sentence (“Based on previous studies, a negative effect of dioxins….”).
Next sentence “TCDD may intensify the pathomechanism of numerous bone diseases, which justifies the search for antagonist substances.”, what pathomechanism and what bone diseases?
Sentence “The purpose of the carried-out experiment is to determine the effect of xantohumol on bone tissue” until to this point, may have failed to me, but I still haven't seen any justification for search the constitution of ions in the bones after exposure to TCDD.
In the end of the 1st paragraph from discussion have the make reference to a study that enounces the possible negative effects of TCDD “reproduction and developmental defects, immunotoxicity, thymus atrophy, epithelial disorders, endocrine imbalance, altered intermediary metabolism, cancer, and wasting syndrome (Jämsä T., Viluksela M., Tuomisto J.T., Tuomisto J., Tuukkanen J.: Effects of 2,3,7,8-Tetrachlorodibenzo-p-dioxin on Bone in Two Rat Strains with Different Aryl Hydrocarbon Receptor Structures. J. Bone Miner. Res. 2001, 16 (10), 1812-1820.)”. Why only here and why only the reference to “decrease of bone mechanical strength and causes an increase of density of cortical bone”?
The discussion is very confusing, for example, I don’t find (in section 3) the data that support this affirmation “the dose of 2 μg / kg b.w. in the group nr VII causes a significant reduction in the calcium content in the bone of cranial vault of the tested Japanese quail.”. And after read the first conclusion.
Please add a reference to this paragraph “Previous reports confirm the need to search for substances that eliminate or reduce the effect of dioxins on bone tissue.”
This phrase is introduction section: “Xanthohumol characterized by the above-described properties can neutralize the effect of dioxins potentiating irregularities in bone tissue related with osteoporosis. In the study of Gao et al. [9], XN has been shown to suppress the activity of T lymphocyte proliferation and cytokine production by Th1 lymphocytes (IL-2, IFN-γ and TNF-α). Activated T cells and the cytokines activated by them lead to RANKL expression on osteoblasts. In addition, stimulated T lymphocytes directly produce RANKL, which by activating a specific RANK receptor, induces the formation and activation of osteoclasts [20]. In addition, it has been observed that xanthohumol can stimulate osteoblast differentiation, induce alkaline phosphatase (ALP) activity and increase marker gene expression of RUNX2 osteoblasts in mouse MC3T3-E116 cells [21].”
Conclusion section.
The conclusion it is a description of results and almost no connection whit discussion section. It has to be highlighted the main conclusions and the novelty of the work in a concise way.
Author Response
Reviewer: 1
General comments:
The manuscript handles with the protective effect of xanthohumol on biological effects caused 2,3,7,8-tetrachlorodibenzo-p-dioxin (TCDD) in bone tissue, although contains some questions.
Detailed comments:
Q1. The introduction has to be concise and objective, but in this case, it starts with one objective, that is repeated 5 phrases after, which is a little strange for those who read. The introduction should give a framework of what is intended to be done and only at the end is the set the objective. In this work, the introduction section, seems to me, a set of random phrases with topics that will be covered and shorter than abstract. In my opinion, have to be reformulated and extended to cover all aspects that will be developed. For example, what kind of negative effects of TCDD can be promoted on bone tissue? What are the probable mechanism involved to decrease or eliminate these negative effects? Nobody else tried to try other compounds in order to get the same result and why xantohumol? Why were measured the concentrations “of calcium (Ca), phosphorus (P), magnesium (Mg), zinc (Zn) and iron (Fe)”?
Answer 1: We have modified the introduction section to be more consistent and have no unnecessary repeats. We have also explained key aspects of bones mechanics and disruption during bones formation.
Q2. In all figures change decimal divisor (,) by (.)
Answer 2: All figures have been corrected.
Q3. It needs references for the first sentence (“Based on previous studies, a negative effect of dioxins….”).
Answer 3: References have been completed.
Q4. Next sentence “TCDD may intensify the pathomechanism of numerous bone diseases, which justifies the search for antagonist substances.”, what pathomechanism and what bone diseases?
Answer 4: The pathomechanism of action of dioxins was briefly described in the introduction.
Q5. Sentence “The purpose of the carried-out experiment is to determine the effect of xantohumol on bone tissue” until to this point, may have failed to me, but I still haven't seen any justification for search the constitution of ions in the bones after exposure to TCDD.
Answer 5: We have modified the aim of the study.
Q6. In the end of the 1st paragraph from discussion have the make reference to a study that enounces the possible negative effects of TCDD “reproduction and developmental defects, immunotoxicity, thymus atrophy, epithelial disorders, endocrine imbalance, altered intermediary metabolism, cancer, and wasting syndrome (Jämsä T., Viluksela M., Tuomisto J.T., Tuomisto J., Tuukkanen J.: Effects of 2,3,7,8-Tetrachlorodibenzo-p-dioxin on Bone in Two Rat Strains with Different Aryl Hydrocarbon Receptor Structures. J. Bone Miner. Res. 2001, 16 (10), 1812-1820.)”. Why only here and why only the reference to “decrease of bone mechanical strength and causes an increase of density of cortical bone”?
Answer 6: We have corrected the bibliography and citations at this part of text, we have found more references that in our opinion are suitable to put them in this place.
Q7. The discussion is very confusing, for example, I don’t find (in section 3) the data that support this affirmation “the dose of 2 μg / kg b.w. in the group nr VII causes a significant reduction in the calcium content in the bone of cranial vault of the tested Japanese quail.”. And after read the first conclusion.
Answer 7: We have rewrite the discussion section and therefore we hope now it is more consistent and understandable.
Q8. Please add a reference to this paragraph “Previous reports confirm the need to search for substances that eliminate or reduce the effect of dioxins on bone tissue.”
Answer 8: References to this paragraph have been completed.
Q9. This phrase is introduction section: “Xanthohumol characterized by the above-described properties can neutralize the effect of dioxins potentiating irregularities in bone tissue related with osteoporosis. In the study of Gao et al. [9], XN has been shown to suppress the activity of T lymphocyte proliferation and cytokine production by Th1 lymphocytes (IL-2, IFN-γ and TNF-α). Activated T cells and the cytokines activated by them lead to RANKL expression on osteoblasts. In addition, stimulated T lymphocytes directly produce RANKL, which by activating a specific RANK receptor, induces the formation and activation of osteoclasts [20]. In addition, it has been observed that xanthohumol can stimulate osteoblast differentiation, induce alkaline phosphatase (ALP) activity and increase marker gene expression of RUNX2 osteoblasts in mouse MC3T3-E116 cells [21].”
Answer 9: This phrase was transferred to the discussion.
Q10. The conclusion it is a description of results and almost no connection whit discussion section. It has to be highlighted the main conclusions and the novelty of the work in a concise way.
Answer 10: The conclusion section have been edited and shortened accordingly to emphasize the main conclusions and the novelty of this paper.
Reviewer 2 Report
The work is potentially interesting, however, in the present form is not suitable for publication.
Apart from the english format, the manuscript has several weaknesses.
introduction : does not suitably present the state of the art, while the first part of the discussion is very focused on data. Perhaps, a reorganization would be appropriate
Table 1 is quite naive.. there is no need to confuse the reader listing the identification number of the single animals.. unless results obtained are discussed animal by animal
If xanthohumol has anti-inflammatory and antioxidant effects, why not use a proven anti-inflammatory and antioxidant drug as a positive control?
I also have some doubts about the results: since there are large deviations for so tiny changes in means, how can be significant the data?
Discussion is not really discussing the data... For example, how to explain the fact that xanthohumol has a hormetic effect?
Author Response
Reviewer: 2
General comments:
The work is potentially interesting, however, in the present form is not suitable for publication. Apart from the english format, the manuscript has several weaknesses.
Detailed comments:
Q1. Introduction : does not suitably present the state of the art, while the first part of the discussion is very focused on data. Perhaps, a reorganization would be appropriate.
Answer 1: The introduction and discussion have been accordingly edited.
Q2. Table 1 is quite naive.. there is no need to confuse the reader listing the identification number of the single animals.. unless results obtained are discussed animal by animal
Answer 2: The table 1 have been corrected.
Q3. If xanthohumol has anti-inflammatory and antioxidant effects, why not use a proven anti-inflammatory and antioxidant drug as a positive control?
Q4. I also have some doubts about the results: since there are large deviations for so tiny changes in means, how can be significant the data?
Q5. Discussion is not really discussing the data... For example, how to explain the fact that xanthohumol has a hormetic effect?
Answer 5: We have rearranged the discussion section and we hope it is now more consistent and understandable. In our study we focused on changes in bones, not in hormone metabolism. We will focus on the hermetic effect of xanthohumol in the future studies.
Reviewer 3 Report
The manuscript "The protective effect of xanthohumol on the content of selected elements in the bone tissue for exposed Japanese quails to TCDD" by Aleksandra Całkosińska and collaborators describes the effect of plant-based xanthohumol in the bone (cranial and hind limb) of quails.The osteoprotective effects of xanthohumol were tested against the dioxin TCDD. Xanthohumol is hypothesised to protect the bone tissue or contribute to bone healing after assault.
The manuscript, in general, requires extensive proofreading and/or English editing.
- The introduction is short and well-structured but it can benefit from including more reports (e.g. about TCDD sources of exposure and current regulations; xanthohumol known pharmacological effects in inflammation and bone healing/ remodeling).
- The methodology is sound and easy to understand. However, other assays should have been performed for a stronger discussion/conclusion (eg. histology/IHC staining; mechanical testing; hydroxyapatite SEM; ELISA). Company /country of spectrometer and other devices used should be added. When determining mineral composition, mention which ions if not total, eg. Ca2+, Fe3+, Zn2+. Could serum markers have been assessed? (eg. osteocalcin,...)
- Results:
The results are very much just descriptive and in a very plain way. English need to be corrected.
in English, you don't use " , " for decimals, you use a dot; for example, it's not p=0,597 but rather p=0.597.
The graphs are adequate and the axis are labelled. All decimals throughout the paper need to be proofread. Is the "constant" SD reported real, accurately calculated?
Repetition: " There is also a statistically significant correlation between TCDD and xanthohumol. The content of this element in the bone of the hind limb significantly depends on the dose of dioxin. There is also a statistically significant correlation between TCDD and xanthohumol" - this happens at 3.4 zinc but also at 3.4. iron ... authors should pay attention to the excessive use of "copy paste" shortcuts!
This copy-paste is also widely used in the figure legends, which can in fact benefit from better descriptions that include TCDD and xanthohumol doses/units indication as well as data transformation applied and reference to relevant p values.
The SD bars seem very constant in all figures and between figures, can the authors clarify? Are the determined mineral contents that constant among the birds of each group as well as between groups?
This report would greatly benefit from extra data, namely IHC of bone tissues. Could you have analysed the hydroxyapatite crystals in the bones and/or bone strength/resistance via mechanical assays?
- Discussion:
References need to be cited right from the start ("Based on previous studies, a negative effect of dioxins on bone tissue, associated with their pro-inflammatory properties and ability to induce oxidative stress, has been demonstrated" - which previous studies?). "That features of TCDD may intensify the pathomechanism of numerous bone diseases, which justifies the search for antagonist substances." - we are left unclear why antagonists are needed and what pathomechanisms are these! specially when TCDD based herbicides and pesticides have been prohibited for decades now.
"The assessment of phosphorus, calcium, magnesium, zinc and iron content makes it possible to determine the effectiveness of bone tissue mineralization processes in tested animals." ; "Previous experimental studies have proved the adverse effect of dioxins..." - Needs reference(s)!
"... group nr IV..." we need to go back to materials&methods to understand which animals were these; please write it here, not just 'group 1, 2 or 3,etc!
"Previous reports confirm the need to search for substances that eliminate or reduce the effect of dioxins on bone tissue." - Refs!?!
"Xanthohumol characterized by the above-described properties can neutralize the effect of dioxins potentiating irregularities in bone tissue related with osteoporosis." - I think this is your hypothesis and not a conclusion?! Define XN.
Then you start to discuss immunology pathways although you did not look into them at all. We are left to question where did this came from all of the sudden and also what do you want to pinpoint here?
"... in the group nr V and VI led to..." which ones are these?
Why did 10mg/kg of XN increased cranial Ca content but not 20mg/kg and nor in the limb?
why do you think it limits "demineralization"?
You did not clarify why increasing XN reduced calcium levels recorded. Nor how calcium content increase with the increase of higher dose of TCDD (aka toxic exposure). Also, why increasing XN usually resulted in the decrease of these elements...?
- Conclusion:
The conclusion is rather long but nicely divided in numbered points. Most of the information here should be in the discussion.
Could be shortened to just state conclusions!
For some reason, here, the authors always use TCDD full name instead of the acronym,...
The analysis reported here is important. However, I find it can be strengthened by further data acquisition.
Author Response
Reviewer: 3
General comments:
The manuscript "The protective effect of xanthohumol on the content of selected elements in the bone tissue for exposed Japanese quails to TCDD" by Aleksandra Całkosińska and collaborators describes the effect of plant-based xanthohumol in the bone (cranial and hind limb) of quails.The osteoprotective effects of xanthohumol were tested against the dioxin TCDD. Xanthohumol is hypothesised to protect the bone tissue or contribute to bone healing after assault.
Detailed comments:
Q1. The manuscript, in general, requires extensive proofreading and/or English editing.
Answer 1: The language of the manuscript were edited.
Q2. The introduction is short and well-structured but it can benefit from including more reports (e.g. about TCDD sources of exposure and current regulations; xanthohumol known pharmacological effects in inflammation and bone healing/ remodeling).
Answer 2: Sources of the exposure to TCDD and the influence of xanthohumol on the inflammation and bone healing/ remodeling was included in the introduction.
Q3. The methodology is sound and easy to understand. However, other assays should have been performed for a stronger discussion/conclusion (eg. histology/IHC staining; mechanical testing; hydroxyapatite SEM; ELISA).
Answer 3: The protocol of this experiment did not provide histological studies as the aim of experiment was to determine the elemental composition of the bone and the amount of tissue material was limited. We appreciate this advise and we will follow them in the future research.
Q4. Company /country of spectrometer and other devices used should be added.
Answer 4: The analysis was performed in acetylene/oxygen flame by means of absorption atomic spectrometry with the use of atomic absorption spectrometer SpectraAA with the
attachment used for work in flame AA240FS (Varian). The measurements of magnesium, iron, manganese, copper and zinc were performed according to the norm PN-EN 14084:2004.
Q5. When determining mineral composition, mention which ions if not total, eg. Ca2+, Fe3+, Zn2+.
Answer 5: We have corrected the whole manuscript and we tried to clarify all phrases.
Q6. Could serum markers have been assessed? (eg. osteocalcin,...)
Answer 6: Our research also provided for collecting serum. Analysis of the composition of the compounds contained in it, however, was not the subject of this study, so the results have not been described. We have not yet collected the complete serum results, therefore we do not refer to them.
Q7. English need to be corrected. In English, you don't use " , " for decimals, you use a dot; for example, it's not p=0,597 but rather p=0.597. All decimals throughout the paper need to be proofread.
Answer 7: All decimals throughout the paper have been corrected.
Q8. Is the "constant" SD reported real, accurately calculated?
Answer 8: We have performed and double-checked all necessary statistical analysis, so the values are real.
Q9. Repetition: " There is also a statistically significant correlation between TCDD and xanthohumol. The content of this element in the bone of the hind limb significantly depends on the dose of dioxin. There is also a statistically significant correlation between TCDD and xanthohumol" - this happens at 3.4 zinc but also at 3.4. iron ... authors should pay attention to the excessive use of "copy paste" shortcuts!
Answer 9: This part of the paper have been corrected.
Q10. This copy-paste is also widely used in the figure legends, which can in fact benefit from better descriptions that include TCDD and xanthohumol doses/units indication as well as data transformation applied and reference to relevant p values.
Answer 10: We have checked and corrected the whole manuscript.
Q11. The SD bars seem very constant in all figures and between figures, can the authors clarify?
Answer 11: The statistical analysis is correct and has been made by the qualified analyst. Standard deviation is a measure of the dispersion of the variable,in our case, i.e. the level of the concentration of elements in the bones of tested individuals. These concentrations depend on the individual properties of the quails studied. Therefore, among others raw results of the concentrations were log transformed.
Q12. Are the determined mineral contents that constant among the birds of each group as well as between groups?
Answer 12: We obtained the results collectively for birds from one group, not individually, so we are not able to answer this question. We do not have such data.
Q13. This report would greatly benefit from extra data, namely IHC of bone tissues. Could you have analysed the hydroxyapatite crystals in the bones and/or bone strength/resistance via mechanical assays?
Answer 13: Unfortunately the amount of tissue material was limited and there was no possibility to perform analysis of hydroxyapatite crystals in the bones and/or bone strength/resistance via mechanical assays.
Q14. References need to be cited right from the start ("Based on previous studies, a negative effect of dioxins on bone tissue, associated with their pro-inflammatory properties and ability to induce oxidative stress, has been demonstrated" - which previous studies?). "That features of TCDD may intensify the pathomechanism of numerous bone diseases, which justifies the search for antagonist substances."
Answer 14: References to the part have been completed.
Q15. We are left unclear why antagonists are needed and what pathomechanisms are these! specially when TCDD based herbicides and pesticides have been prohibited for decades now.
Answer 15: Dioxins belong to the group of persistent organic compounds and accumulate in the environment at every stage of the food chain. It leads to their accumulation in the adipose tissue of organisms and permanent exposure to low doses of dioxins, which may result in the occurrence of various types of health disorders. In places where ecological disasters have occurred (eg. Seveso (Italy)), the population is particularly exposed to the effects of dioxins. This justifies the search for pharmacological agents that have protective effect against dioxins.
Q16. "The assessment of phosphorus, calcium, magnesium, zinc and iron content makes it possible to determine the effectiveness of bone tissue mineralization processes in tested animals." ; "Previous experimental studies have proved the adverse effect of dioxins..." - Needs reference(s)!
Answer 16: References to the part have been completed.
Q17. "... group nr IV..." we need to go back to materials&methods to understand which animals were these; please write it here, not just 'group 1, 2 or 3,etc!
Answer 17: This part of the manuscript have been edited.
Q18. "Previous reports confirm the need to search for substances that eliminate or reduce the effect of dioxins on bone tissue." - Refs!?!
Answer 18: References to the part have been completed.
Q19. "Xanthohumol characterized by the above-described properties can neutralize the effect of dioxins potentiating irregularities in bone tissue related with osteoporosis." - I think this is your hypothesis and not a conclusion?! Define XN.
Answer 19: This paragraph of the manuscript have been edited.
Q20. Then you start to discuss immunology pathways although you did not look into them at all. We are left to question where did this came from all of the sudden and also what do you want to pinpoint here?
Answer 20: This part of the manuscript have been edited.
Q21. "... in the group nr V and VI led to..." which ones are these?
Answer 21: This paragraph of the manuscript have been edited and the precise dosages of the substances used have been determined.
Q22. Why did 10mg/kg of XN increased cranial Ca content but not 20mg/kg and nor in the limb?
why do you think it limits "demineralization"?
Answer 22: We have explained the differences in bone types within the manuscript and we hope the explanation is clear.
Q23. You did not clarify why increasing XN reduced calcium levels recorded. Nor how calcium content increase with the increase of higher dose of TCDD (aka toxic exposure). Also, why increasing XN usually resulted in the decrease of these elements...?
Q24. The conclusion is rather long but nicely divided in numbered points. Most of the information here should be in the discussion. Could be shortened to just state conclusions!
Answer 24: The conclusion have been edited and shortened accordingly to emphasize the main conclusions and the novelty of this paper.
Q25. Conclusion: For some reason, here, the authors always use TCDD full name instead of the acronym,...
Answer 25: This part of the manuscript has been edited.
Q26. The analysis reported here is important. However, I find it can be strengthened by further data acquisition.
Answer 26: Thank you for your kind words. In the future, we will try to expand research on this topic, also considering it as important.
Round 2
Reviewer 1 Report
The scientific article has been improved in its generality.
Author Response
Dear Reviewer #1,
We are very grateful for your opinion, which contributed to the improvement of the quality of the submitted manuscript.
Kind regards,
Aleksandra Całkosińska (DDS, PhD).
Reviewer 2 Report
ok, all major problems have been corrected. Please check editing.
Author Response
Dear Reviewer #2,
we have carefully read the whole paper and corrected mistakes that we have found in it. We hope there are no more mistakes, typing errors or any other editing issues. Thank you very much for your kind decision.
Kind regards,
Aleksandra Całkosińska (DDS, PhD).
Reviewer 3 Report
The manuscript entitled "The protective effect of xanthohumol on the content of selected elements in the bone tissue for exposed Japanese quails to TCDD" significantly improved after authors reviewing.
However, the manuscript still requires proofreading throughout. Examples:
Abstract -
... morphological and functional abnormalities including in the bone tissue ... The aim of this study was to assess ...After euthanasia of the animals,...zinc (Zn) and iron (Fe). The results of the experiment... ...dioxins. These results confirm...
Moreover, the abstract does not include a small description of the results or highlights of the experimental work; just final conclusions are mentioned.
Introduction
"This research was undertaken to determine the effect of xanthohumol against the harmful effects caused by 2,3,7,8-tetrachlorodibenzo-p-dioxin (TCDD) in the bone tissue." etc etc Introduction improved significantly now giving a reasonable background for the proposed work. To note here, though, that the information about XN needs to be rearranged in order to make the introduction flow properly. eg The authors talk about the role of XN as phytoestrogen before introducing its Latin name and where it is extracted from... Materials and Methods & Results Subtitle "Animals" is not present. Table and figure legends needs improvement. E.g. : "Table 1. Scheme of the animal groups formed for the experiment. The animals were divided in nine groups, including a negative control group. TCDD and Xanthohumol were administered "intramuscularly to birds in different doses."Figure 1a. Average calcium concentration (transformed variables) in the bone of the cranial vault depending on the dose of TCDD and xanthohumol and results of the significance test. Results are represented as mean +- SD. P<0.05 was considered statistically significant. Significant p value is highlighted in bold."
Font in figures also needs to increase for readability!
I believe it would be positive to include in the manuscript (namely as a background for the authors' objectives with this experiment) the following explanation for persistent organic pollutants given by the authors:
- "Dioxins belong to the group of persistent organic compounds and accumulate in the environment at every stage of the food chain. It leads to their accumulation in the adipose tissue of organisms and permanent exposure to low doses of dioxins, which may result in the occurrence of various types of health disorders. In places where ecological disasters have occurred (eg. Seveso (Italy)), the population is particularly exposed to the effects of dioxins. This justifies the search for pharmacological agents that have protective effect against dioxins.".
I am still left unclear exactly what forms of the minerals were measured.
The work downfall lies on being based in a single experiment. No other methods are used to test the authors hypothesis (XN protective or regenerative effects on bone damage caused by TCDD). The scientific plan for this hypothesis should have included techniques as micro-computed tomography, histopathology, serum biochemistry (namely for biomarkers of bone turnover), just to name a few. The low material amount argument when using quails does not sound right.
However, the results reported here should be significant for the field, and, similar investigation was not found in the literature.
Author Response
Dear Reviewer #3,
We would like you to express our sincerest gratitude for this feedback. Point-by-point answers to your remarks are provided below.
Detailed comments:
Q1. Moreover, the abstract does not include a small description of the results or highlights of the experimental work; just final conclusions are mentioned.
Answer 1: We have revised the Abstract and include description of methods and a short description of results.
Q2. Manuscript still requires proofreading throughout.
Answer 2: We have verified the whole manuscript and corrected errors within the text, figures and table.
Q3. Introduction improved significantly now giving a reasonable background for the proposed work. To note here, though, that the information about XN needs to be rearranged in order to make the introduction flow properly. eg The authors talk about the role of XN as phytoestrogen before introducing its Latin name and where it is extracted from...
Answer 3: The introduction was rearranged in order to achieve the proper flow of information.
Q4. Subtitle "Animals" is not present.
Answer 4: Subtitle „Animals and experimental groups” has been supplemented.
Q5. Table and figure legends needs improvement. E.g. : "Table 1. Scheme of the animal groups formed for the experiment. The animals were divided in nine groups, including a negative control group. TCDD and Xanthohumol were administered "intramuscularly to birds in different doses." Figure 1a. Average calcium concentration (transformed variables) in the bone of the cranial vault depending on the dose of TCDD and xanthohumol and results of the significance test. Results are represented as mean +- SD. P<0.05 was considered statistically significant. Significant p value is highlighted in bold."
Answer 5: Table and figure legends have been revised as recommended.
Q6. Font in figures also needs to increase for readability!
Answer 6: The font in figures was improved to achieve better readability. We have sent .tif files that should provide good quality of figures.
Q7. I believe it would be positive to include in the manuscript (namely as a background for the authors' objectives with this experiment) the following explanation for persistent organic pollutants given by the authors:
"Dioxins belong to the group of persistent organic compounds and accumulate in the environment at every stage of the food chain. It leads to their accumulation in the adipose tissue of organisms and permanent exposure to low doses of dioxins, which may result in the occurrence of various types of health disorders. In places where ecological disasters have occurred (eg. Seveso (Italy)), the population is particularly exposed to the effects of dioxins. This justifies the search for pharmacological agents that have protective effect against dioxins.".
Answer 7: The following explanation for persistent organic pollutants has been supplemented in the Introduction.
Q8. I am still left unclear exactly what forms of the minerals were measured.
Answer 8: The analyte in our experiment consists of homogenized and wet mineralized. Then samples were transferred into vessels and the concentration of elements were measured using emission atomic spectrometry method and absorption atomic spectrometry method.
Q9. The work downfall lies on being based in a single experiment. No other methods are used to test the authors hypothesis (XN protective or regenerative effects on bone damage caused by TCDD). The scientific plan for this hypothesis should have included techniques as micro-computed tomography, histopathology, serum biochemistry (namely for biomarkers of bone turnover), just to name a few. The low material amount argument when using quails does not sound right.
However, the results reported here should be significant for the field, and, similar investigation was not found in the literature.
Answer 9: It is true that the work is based on one study, but the experience model is innovative and unique. Until now, similar studies were carried out by us on the rat and chicken models. (DOI: 10.1007/s12011-018-1580-y and DOI: 10.1039/C8RA10485A)
When planning further work, we will certainly take your suggestions into account, because they are extremely valuable and will certainly improve the substantive value of subsequent experiences.
Unfortunately, now we are not able to carry out the additional tests suggested by you, because the material obtained from quails was used in above described experiment.
However, we hope that the originality and the amount of work put into conducting the experiment will affect the issuance of a positive assessment.
Kind regards,
Aleksandra Całkosińska (DDS, PhD).